# Study of Personal Mobility Vehicle (PMV) with Active Inward Tilting Mechanism on Obstacle Avoidance and Energy Efficiency

**Tetsunori Haraguchi [1,\*], Ichiro Kageyama [1] and Tetsuya Kaneko [2]**

[1] College of Industrial Technology, Nihon University, 1-2-1 Izumi-cho, Narashino, Chiba 275-8575, Japan; kageyama.ichiro@nihon-u.ac.jp

[2] Department of Mechanical Engineering for Transportation, Osaka Sangyo University, 3-1-1 Nakagaito, Daito 574-8530, Japan; kaneko@tm.osaka-sandai.ac.jp

\* Correspondence: haraguchi@nagoya-u.jp; Tel.: +81-47-474-2337



**Featured Application: CarMaker of IPG Automotive in Germany was used as a vehicle dynamics simulation tool.**

**Abstract:** Traffic congestion and lack of parking spaces in urban areas etc. may lessen the original benefits of cars, and now new ultra-small mobility concepts called personal mobility vehicles (PMVs) are receiving attention. Among them, PMVs with an inward tilting mechanism in order to avoid overturning on turning, as in motorcycles, look realistic in new, innovative traffic systems. In this study, PMVs with an active inward tilting mechanism with three wheels, double front wheels and single rear wheel, were studied regarding front inner wheel lifting phenomena, capability of obstacle avoidance and energy balance of active tilting mechanism. Based on a comprehensive study of inner wheel lifting and obstacle avoidance, PMVs with a front wheel steering system as the most realistic specification were compared on capability of obstacle avoidance with passenger cars and motorcycles with those that have current market experience, and showed better capability. Although the energy consumption of an active inward tilting mechanism might be in conflict with the energy efficiency of small PMV concepts, the energy needed to tilt PMVs was very little compared with the general energy consumption of driving. It was clarified that the new PMV concepts with inward tilting mechanism have sufficient social acceptability from both mandatory points of safety and efficiency.

**Keywords:** personal mobility vehicle; active tilting; inner wheel lifting; obstacle avoidance; energy efficiency

## 1. Introduction

Although automobiles have improved the quality of life of human beings, too big an increase in the number of vehicles in use has the potential to spoil the original benefits of automobiles, due to problems such as traffic congestion and lack of parking space in urban areas. Thus, new ultra-small mobility concepts called personal mobility vehicles (PMVs) are now attracting attention [1,2].

PMVs with narrow total width are likely to be of the inward tilting type, the same as motorcycles, in order to avoid overturning on turning. Among them, tricycles with two front wheels and one rear wheel seem to be a good idea because of the simplicity of the vehicle configuration and security against overturning during braking, as shown in Figure 1. A general understanding of the advantages and disadvantages of the different types of PMV are shown in Table 1.

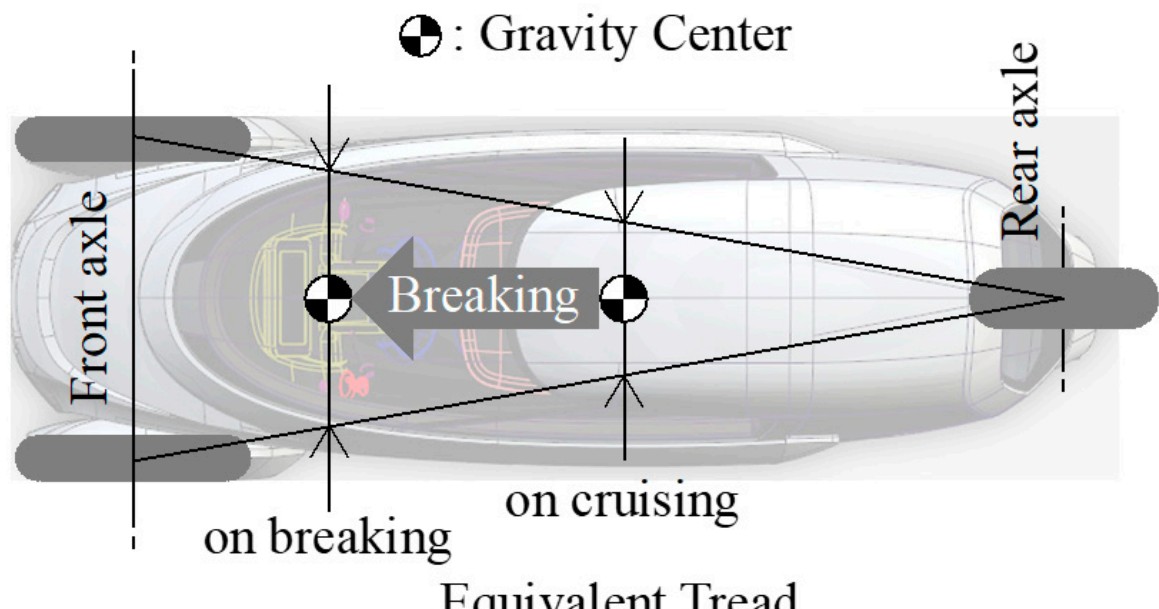

**Figure 1.** Equivalent tread is wider on breaking.

**Table 1.** Advantages (✔)/Disadvantages (▼) of different types of personal mobility vehicles (PMVs).

| | | | | Simple Construction | Slow Speed | Stability on Braking | Obstacle Avoidance |
|---|---|---|---|---|---|---|---|
| Single front wheel+Double rear wheels | | | | ✔ | | ▼Unstable | ▼ U.S. |
| | Double front wheels + Single rear wheel | | | | | ✔ | |
| | | Front steering + Rear traction | | | | | |
| | | | Passive tilting mechanism | ✔ | ▼Falling | | ▼Delay |
| | | | Active tilting mechanism | ▼Active | ✔ | | ✔ |
| | Front traction + Rear steering | | | ▼Steer-by-wire | | | ▼Delay |
| | Double front wheels + Double rear wheels | | | ▼4 wheels | | ✔ | |
| | | Front steering + Rear traction | | | | | |
| | | | Passive tilting mechanism | ✔ | ▼Falling | | ▼Delay |
| | | | Active tilting mechanism | ▼Active | ✔ | | ✔ |

The following concerns should be clarified in order to implement new PMV concepts with inward tilting mechanisms in future mobility society.

The following is the main concern regarding PMVs with a passive tilting mechanism:

- Self-standing ability from stop to very low speed

The following are the main concerns regarding PMVs with an active tilting mechanism:

- Inner wheel lifting on sudden steering input
- Capability on obstacle avoidance
- Energy consumption on active tilting system

*1.1. Self-Standing Ability from Stop to Very Low Speed*

In order to achieve both a passive tilting mechanism and self-standing ability from stop to very low speed, it should be a simple, mechanical mechanism. Although the details are omitted this time, we devised a self-standing mechanism and confirmed the effectiveness of that mechanism using a motorcycle with two front wheels, as shown in Figure 2.

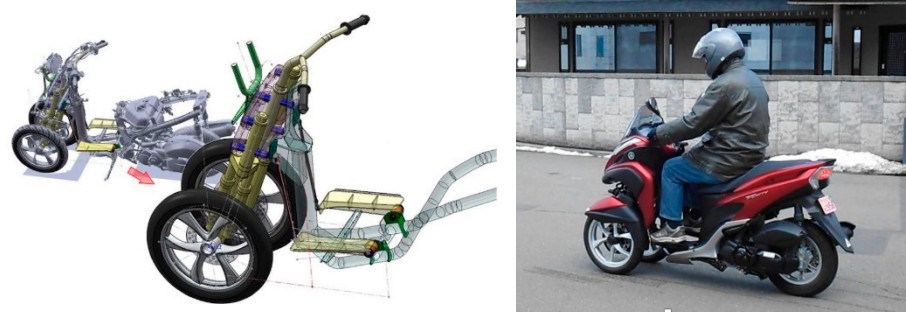

**Figure 2.** Self-standing mechanism was devised.

## 1.2. Study of PMVs with Active Tilting Mechanism

In this report, using the multibody dynamics models described in the next chapter, we study three concerns regarding PMVs with active tilting mechanisms and discuss the social acceptability of inward tilting type PMVs.

## 2. Multibody Dynamics Model

### 2.1. Vehicle Model

The specifications of a PMV are shown in Figure 3 and Table 2. Although overall length and width are about half of that of a passenger car, overall height is similar to a passenger car. Therefore, inward tilting is necessary.

CarMaker [3] of IPG Automotive in Germany was used as a vehicle dynamics simulation tool. In order to construct a tilting model with the car simulation tool, the following measures were taken.

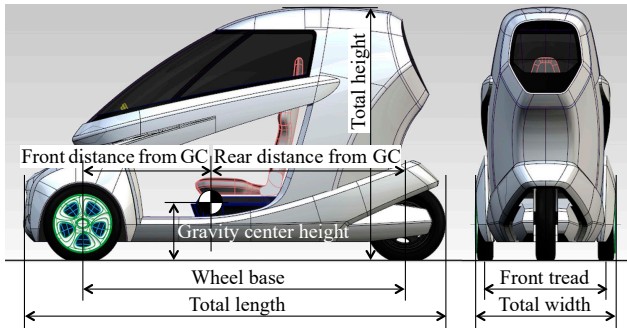

**Figure 3.** Dimensions of model vehicle.

**Table 2.** Specifications of model vehicle.

| Item | Unit | Value | Item | Unit | Value |
|---|---|---|---|---|---|
| Total length | m | 2.645 | Total mass | kg | 369.8 |
| Total width | m | 0.880 | Front mass distribution | kg | 222.1 |
| Total height | m | 1.445 | Rear mass distribution | kg | 147.7 |
| Wheel base | m | 2.020 | Roll moment of inertia | kgm$^2$ | 58.8 |
| Front distance from gravity center | m | 0.807 | (Roll moment of inertia of sprung mass) | kgm$^2$ | 43.0 |
| Rear distance from gravity center | m | 1.213 | Pitch moment of inertia | kgm$^2$ | 197.3 |
| Front tread | m | 0.850 | (Pitch moment of inertia of sprung mass) | kgm$^2$ | 118.0 |
| Gravity center height | m | 0.358 | Yaw moment of inertia | kgm$^2$ | 187.3 |
| Steering gear ratio | - | 16.0 | (Yaw moment of inertia of sprung mass) | kgm$^2$ | 102.3 |

### 2.1.1. Suspension

As shown in Figure 4, forced torsional torque is applied to the stabilizer bar. In this method, the vehicle generates internal torque. Although the mechanism might be different from PMVs having direct suspension stroke control mechanisms, the motion characteristics and external forces acting on this vehicle are same.

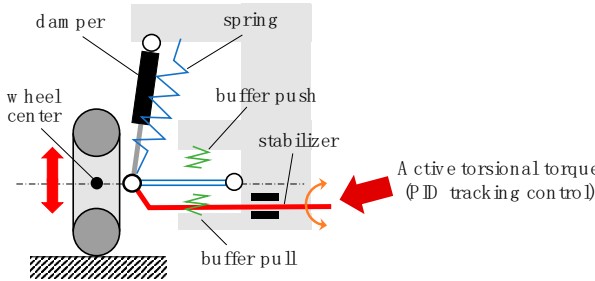

**Figure 4.** Torsional torque is applied to stabilizer bar.

### 2.1.2. Target Roll Angle

Target tilting (roll) angle is given as shown in Equation (1), to balance against virtual lateral acceleration, which is simply calculated from tire steered angles, wheel base and vehicle speed. Torsion torque of the active stabilizer to get the target roll angle is given by general proportional-integral-derivative (PID) tracking control. Initial values of control gain are shown in Equation (2).

$$TRA = A \tan^{-1}\left(\frac{\sin(\delta)v^2}{lg}\right) \tag{1}$$

$$P = 4000, I = 100, D = 0 \tag{2}$$

*TRA*: target roll angle
*A*: user amplification factor
*δ*: tire steered angle
*v*: vehicle speed
*l*: wheel base
P: proportional gain
I: integral gain
D: derivative gain

### 2.1.3. Tire

PMVs with tilting mechanisms are strongly influenced by tire camber characteristics. Therefore, we used a motorcycle tire model in CarMaker as shown in Figure 5 to consider the side force on tire camber angle.

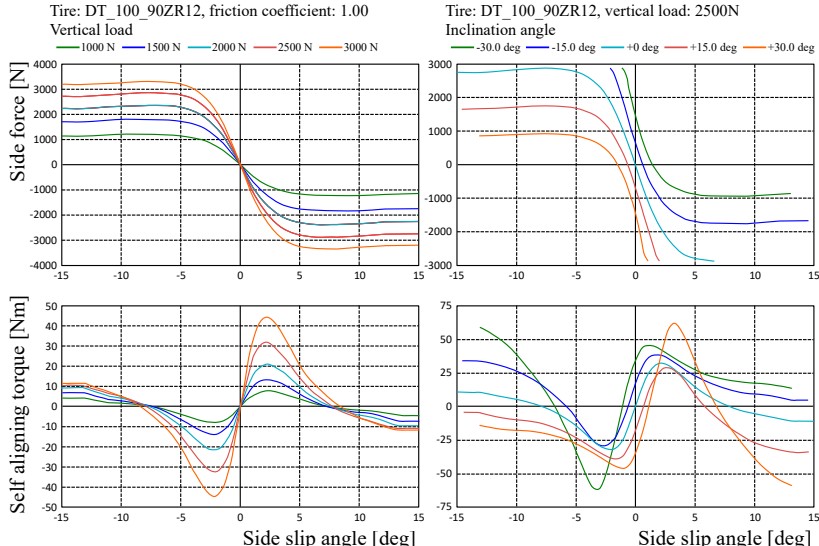

**Figure 5.** Motorcycle tire model is used in CarMaker.

*2.2. Driver Model*

IPGDriver from IPG Automotive attached to CarMaker is used as driver model in this study. IPGDriver understands the vehicle characteristics and controls the vehicle speed and steering wheel angle to trace the given course based on a forward quadratic prediction model.

**3. Inner Wheel Lifting Phenomena**

An active tilting mechanism forcibly applies strokes to suspension. In the case where the roll response of the sprung body is delayed, a difference of vertical loads on the front two wheels occurs. In worse cases, inner wheel lifting phenomena [4,5] occur. Although lifting itself is not a problem, divergent phenomena cause overturning. Therefore, it is better to avoid repeated liftings.

*3.1. Understanding Mechanism of Lifting Phenomena*

The mechanism of front inner wheel lifting caused by sudden steering inputs seems to be as follows.

1. Inward roll response to steering input cannot occur in time due to roll moment of inertia.
2. Transitional front wheel slip angle due to vehicle response delay caused by yaw moment of inertia causes outward roll moment. This roll moment further delays inward roll response.
3. Too much rebound stroke of front outer wheel caused by inward roll response delay lifts up front body. Pitch moment of inertia delays recovery from front body lifting.

3.1.1. Influence of PID Factors on Roll Angle Tracking Control

In order to study the influence of three axes moments of inertia on front inner wheel lifting phenomena during sudden steering inputs, it is inconvenient if the vehicle has unstable roll phenomena (for example, roll vibration) without intentional steering input. As shown in Figure 6, when the roll moment of inertia of sprung mass ($Ixx^*$) is increased, the vehicle becomes unstable in roll direction. Roll vibration starts with doubled $Ixx^*$ even on straight running ahead. Overturning occurs with quadrupled $Ixx^*$ without ability to continue straight running, as shown in Figure 7.

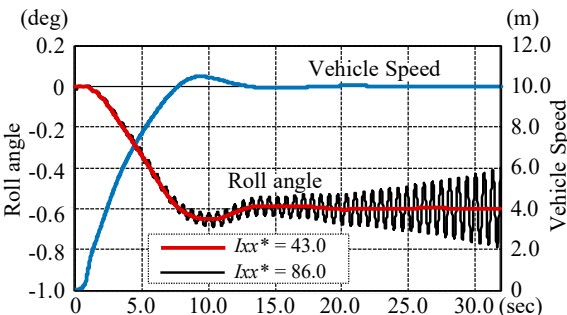

**Figure 6.** Unstable roll vibration with high *Ixx\**.

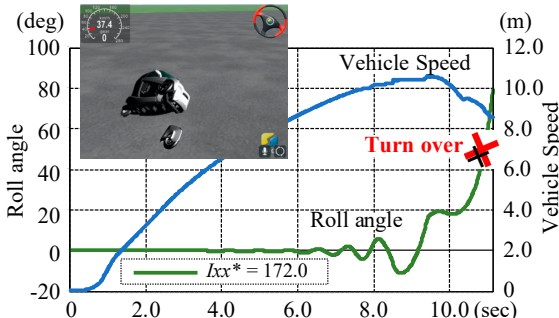

**Figure 7.** Turning over with too high roll moment of inertia of sprung mass (*Ixx\**).

As shown in Figure 8, in case I gain was reduced by half (100→50), it was confirmed that roll vibration is suppressed even under previous conditions (doubled *Ixx\** and quadrupled *Ixx\**). There is no roll vibration even the roll moment of inertia is quadrupled. We proceeded with I gain = 50 in order to eliminate the disturbing effect of roll tracking control.

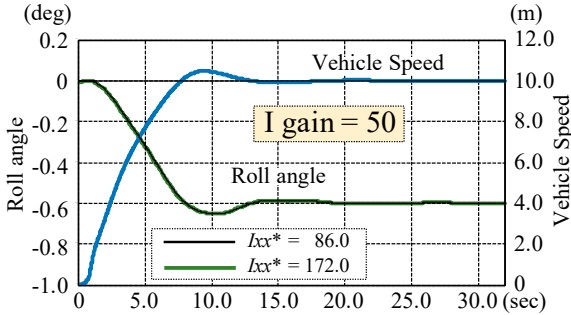

**Figure 8.** No roll vibration phenomena are observed (I = 100→50).

### 3.1.2. Steering Input Condition

The lateral displacement of motorcycles in the severe obstacle avoidance scene was about 2 m. The steering input angle was set as the initial condition to get this lateral displacement. Lateral displacement *LD* is proportional to steering angle *MA* and the square of vehicle speed *v*, and inversely proportional to the square of steering input frequency *f*, as in Equation (3).

Sinusoidal input of about 0.5 Hz ± 60 deg gives about 2 m lateral displacement *LD* at a vehicle speed of 36 km/h (10 m/s). An input frequency of 0.5 Hz is equivalent to quick lane change (L/C) operation of standard drivers on public roads.

$$LD \propto \frac{MA\,v^2}{f^2} \tag{3}$$

*LD*: lateral displacement

*MA*: steering wheel angle

*v*: vehicle velocity

*f*: steering input frequency

### 3.1.3. Relation between Steering Angle and Input Frequency

Lateral displacement *LD* is expressed using Equation (3). Steering input angle *MA* proportioned to the square of steering input frequency *f* gives maintained *LD*, as shown in Figure 9. However, an extreme increase of steering input angle *MA* means fairly severe conditions for inner wheel lifting phenomena. Therefore, increasing steering input frequency *f* results in a sudden turnover.

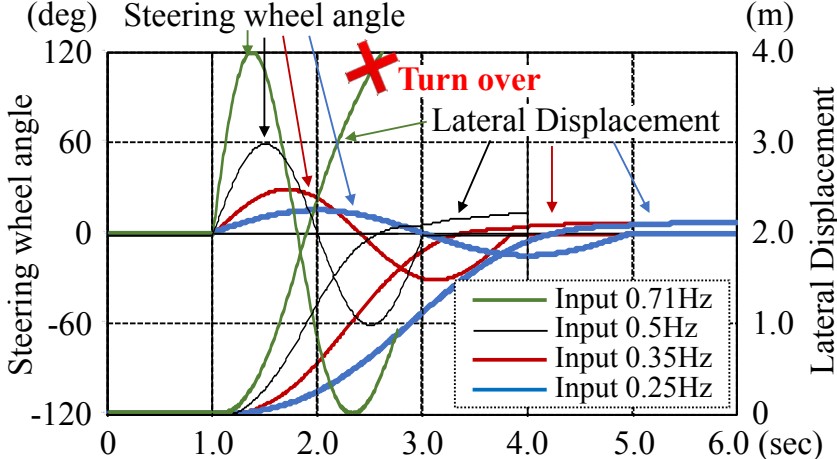

**Figure 9.** Steering angle proportioned to square of frequency results in sudden turnover (I = 50).

Only steering input frequency *f* was changed, while maintaining steering input angle *MA* = ±60 deg, without maintaining *LD*, as shown in Figure 10. The minimum residual load of the front left wheel (inner wheel) is almost zero at *f* = 0.71 Hz, and the inner wheel lifts clearly at *f* = 1.0 Hz. It can be understood that front inner wheel lifting is caused not only by the large steering angle input but also directly by quick steering input.

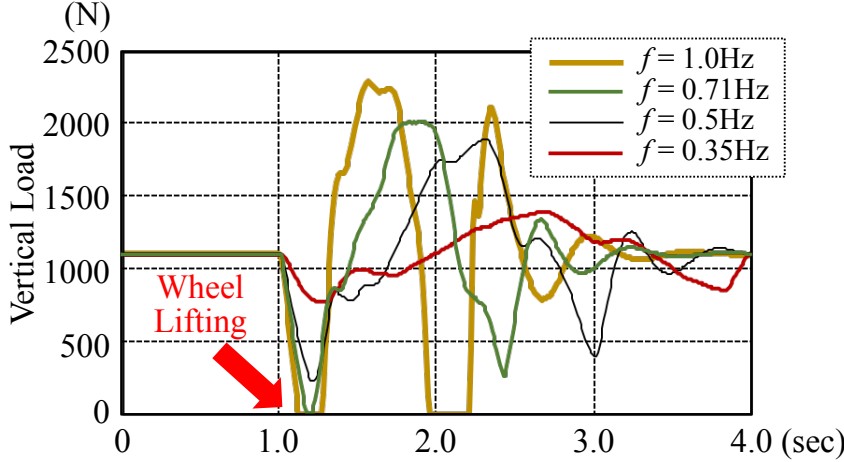

**Figure 10.** Quick steering input directly causes wheel lifting (I = 50).

Quick and large steering input is actually very difficult for human drivers. The effects of three axis moments of inertia on front inner wheel lifting phenomena by sudden steering input were examined with 0.5 Hz (quick L/C) as the upper limit of standard drivers' ability.

### 3.1.4. Vehicle Speed

Steering input angle *MA* inversely proportioned to the square of vehicle speed *v* gives maintained *LD*, as shown in Figure 11. However, in order to avoid obstacles completely, steering input frequency *f* needs to be increased in proportion to the increase of vehicle speed *v*. Thus, *MA* is not decreased proportionally to the square of vehicle speed *v* on actual obstacle avoidance, as shown in Figure 12. The effects of three axis moments of inertia on front inner wheel lifting phenomena by sudden steering input were examined with 36 km/h (10 m/s).

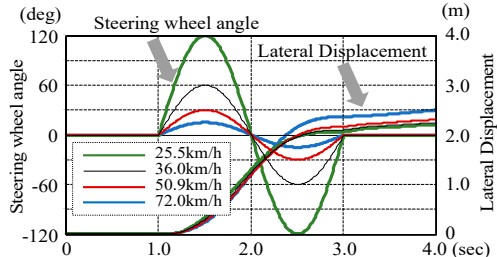

**Figure 11.** Required steering input angle (I = 50).

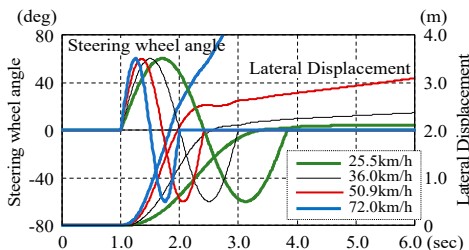

**Figure 12.** Influence of vehicle speed (I = 50).

### 3.2. Influences of Three Axis Moments of Inertia

The effects of moments of inertia of sprung mass around three axes *Ixx\**, *Iyy\** and *Izz\** were examined.

### 3.2.1. Influence of Roll Moment of Inertia

The influence of roll moment of inertia was examined by parameterizing roll moment of inertia of sprung mass *Ixx\** out of whole vehicle roll moment of inertia *Ixx* = 58.8 kg/m$^2$. Figure 13 (I = 50, *f* = 0.5 Hz) shows vehicle behavior at vehicle speed *v* = 36 km/h (10 m/s), steering input frequency *f* = 0.5 Hz, sinusoidal steering angle *MA* = ±60 deg, and roll tracking gain I = 50.

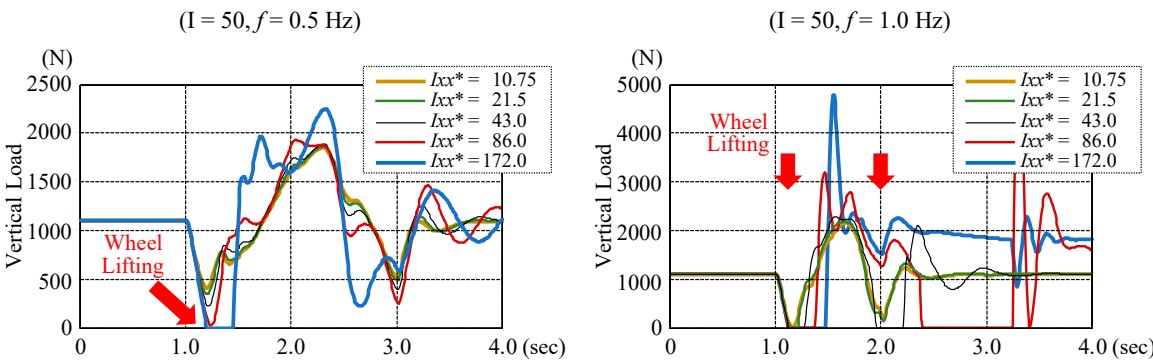

**Figure 13.** Influence of roll moment of inertia on vertical load of front left wheel (I = 50).

Larger *Ixx** clearly increase front inner wheel lifting. Although no phenomena occur when *Ixx** = 43.0 kgm$^2$ (standard), almost zero vertical load of inner wheel occurs when this is doubled to *Ixx** = 86.0 kgm$^2$, and inner wheel lifting phenomenon clearly occurs when this amount is doubled again to *Ixx** = 172.0 kgm$^2$.

The effect of roll moment of inertia of sprung mass *Ixx** was confirmed also at *f* = 1.0 Hz (sudden L/C). As shown in Figure 13 (I = 50, *f* = 1.0 Hz), front inner wheel lifting occurs even when *Ixx** = 43.0 kgm$^2$ (standard).

### 3.2.2. Influence of Pitch and Yaw Moments of Inertia

Influences of pitch moment of inertia of sprung mass *Iyy** and yaw moment of inertia of sprung mass *Izz** were examined as shown in Figure 14. However, no significant influences were observed on front inner wheel lifting phenomena. Therefore, we decided to concentrate on roll direction mechanism 1, as proposed in Section 3.1.

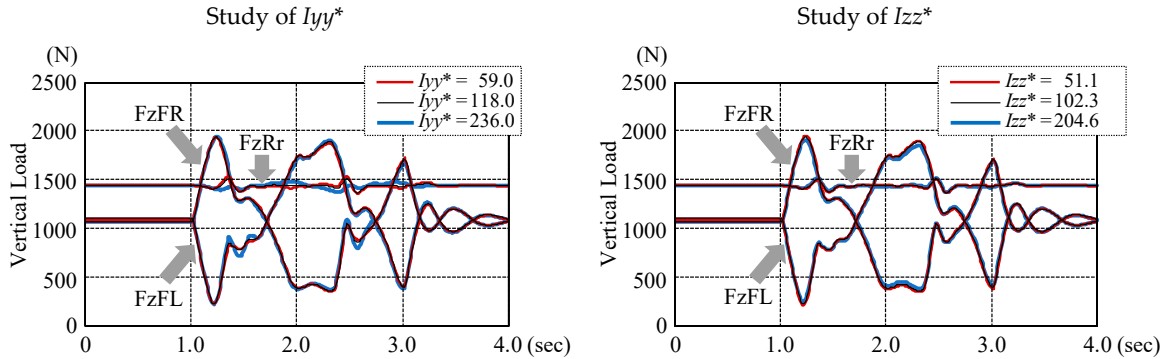

**Figure 14.** No significant influences were observed with pitch and yaw moments of inertia (I = 50).

### 3.2.3. Roll Resonance Frequency *fφ* and Influence of Roll Dynamics

PMVs have half total width and same total height (equivalent to nearly half roll moment of inertia), and half front tread (equivalent to one quarter front roll stiffness) compared with passenger cars. PMVs have no roll stiffness on the rear wheel (equivalent to one eighth total roll stiffness). Thus, as shown in Equation (4), roll resonance frequency *fφ* is judged to be further lower (equivalent to 1/2).

Therefore, we grasp roll resonance frequency *fφ*, experimentally at first. While driving straight at 36 km/h (10 m/s), a roll pulse was input by a sharp steering pulse (±30 deg, 8 Hz, one cycle of sinusoidal input), as shown in Figure 15. After the pulse input, remained roll vibration was observed, and its frequency was about 1.67 Hz. This roll resonance frequency *fφ* is on suspension roll stiffness, not related to the tilting mechanism.

As shown in Figure 16, vehicle mass, spring/stabilizer-bar stiffness of suspension, damping force characteristics, and roll tracking control gains P, I, D were all doubled in order to get vehicle

characteristics with equivalent roll dynamics. Subsequently, front inner wheel lifting phenomena showed almost the same characteristics as the reference vehicle. From this, it was confirmed that front inner wheel lifting phenomena are almost dominated by the roll dynamics of PMVs.

$$f\varphi \propto \left(\frac{K\varphi}{Ixx}\right)^{\frac{1}{2}}$$

(4)

*fφ*: roll resonance frequency

*Kφ*: roll stiffness

*Ixx*: roll moment of inertia

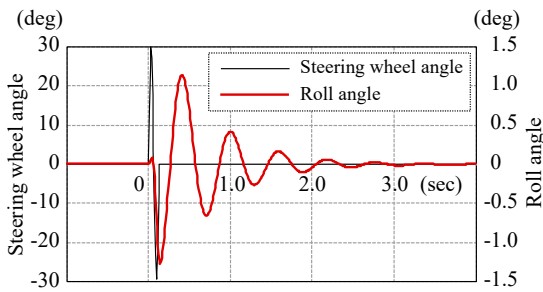

**Figure 15.** Roll resonance frequency is about 1.67 Hz (I = 100).

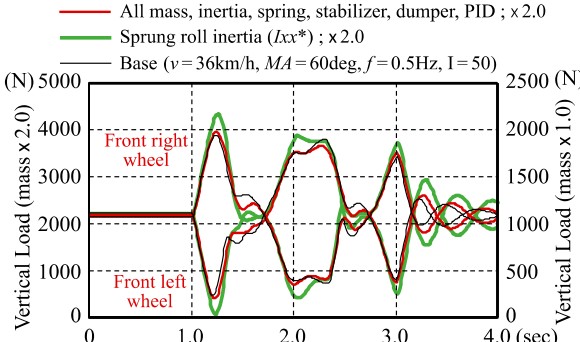

**Figure 16.** Equivalent roll dynamics gives same wheel lifting phenomena.

*3.3. Summary of This Section and Setting Specifications for Further Study*

- Lateral displacement *LD* is proportional to steering angle *MA* and the square of vehicle speed *v*, and inversely proportional to the square of steering input frequency *f*.
- Increase of steering input frequency *f* itself is disadvantageous for front inner wheel lifting. Increase of input steering wheel angle *MA* to maintain lateral displacement *LD* on increase of steering input frequency *f* causes further severe conditions of front inner wheel lifting.
- Increase of vehicle speed *v* requires quicker steering input frequency *f*, which is disadvantageous for front inner wheel lifting.
- Inner wheel lifting phenomenon is only that vehicle roll is unavailable to follow target roll angle *TRA* due to roll moment of inertia of sprung mass *Ixx\** and is almost dominated by the roll dynamics of PMVs. Smaller *Ixx\** and larger *Kφ*, i.e., higher roll resonance frequency *fφ* improves front inner wheel lifting phenomena.
- Unstable roll phenomena without intentional steering input related to front inner wheel lifting phenomena is suppressed by decrease of I gain of PID tracking control to active roll angle. In order to study realistic roll moment of inertia *Ixx\** and roll stiffness *Kφ*, PID control parameters in further study are set as Equation (5).

$$P \; = \; 4000, \; I \; = \; 50, \; D \; = \; 0 \tag{5}$$

## 4. Capability on Obstacle Avoidance

### 4.1. Comparison Front Wheel Steering (FWS) and Rear Wheel Steering (RWS)

Recently, Toyota Motor Corporation introduced an active tilting type PMV concept vehicle with RWS, considering the package advantageous. However, the RWS system is not generally available on the market. A comparison of FWS and RWS from the point of view of obstacle avoidance is necessary [6].

#### 4.1.1. Understanding of Vehicle Posture on Avoidance Behavior

Although there is no difference between FWS and RWS in Equation (6), which describes vehicle condition on steady circle, RWS vehicles show a clearly inward posture (rear end pushed out), as shown in Figure 17.

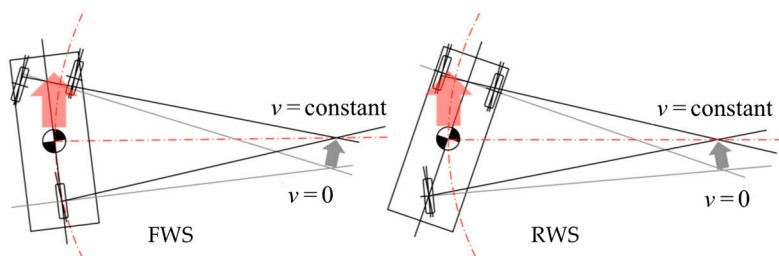

**Figure 17.** Vehicle postures on steady-state cornering.

An image of vehicle posture on left direction avoidance is shown on Figure 18. The rear vehicle end moves to right at first in RWS, and this causes a delay in the lateral displacement *LD* of RWS vehicles.

$$\rho \; = \; \left(1 - \frac{m}{2l^2} \frac{l_f K_f - l_r K_r}{K_f K_r} v^2\right)\frac{l}{\delta} \tag{6}$$

$\rho$: turning radius
$m$: vehicle mass
$l$: wheel base
$l_f$: front wheel base
$K_f$: front cornering power
$l_r$: rear wheel base
$v$: velocity
$\delta$: tire steer angle

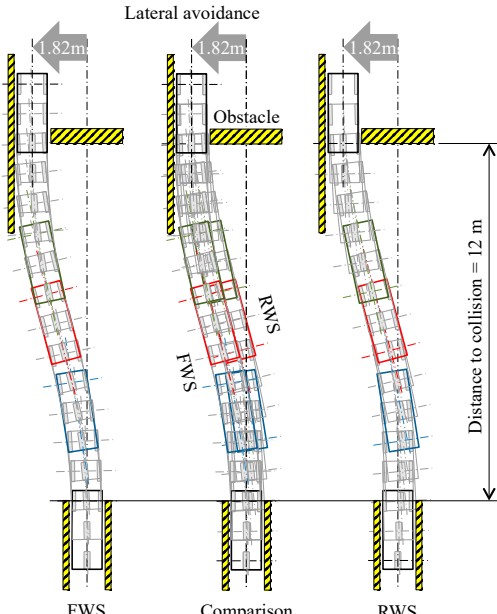

**Figure 18.** Tracks image of front wheel steering (FWS) and rear wheel steering (RWS) to avoid obstacle.

### 4.1.2. Open Loop Simulation

First, open loop simulations were conducted using steering angle input as shown on the left in Figure 19. Vehicle speed was fixed at 36 km/h and lateral displacement was 1.82 m, achieved with sinusoidal steering input. The delay in the lateral displacement *LD* in RWS assumed in the former section is also shown in comparison using a dynamic simulation tool, shown on the right in Figure 19. There is an approximately 2 m delay in longitudinal distance under this condition.

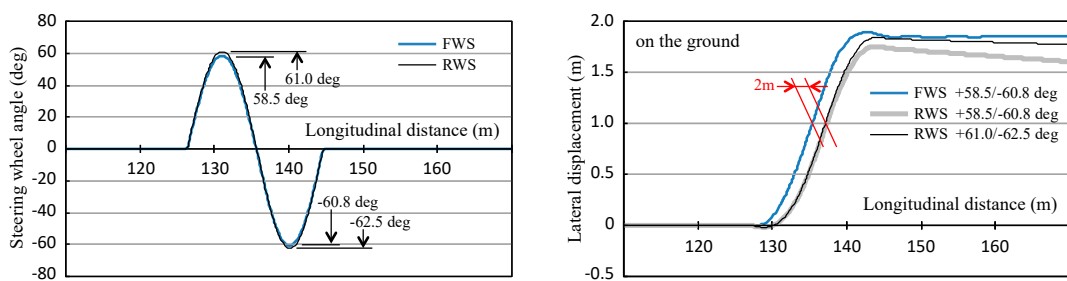

**Figure 19.** Steering wheel angle and lateral displacement on open loop simulation.

### 4.1.3. Closed Loop Simulation

A pylon-controlled single lane change course was set for closed loop simulation, as shown in Figure 20. Maximum steering angle, maximum steering speed, and maximum steering acceleration are all sufficiently large for avoidance operation.

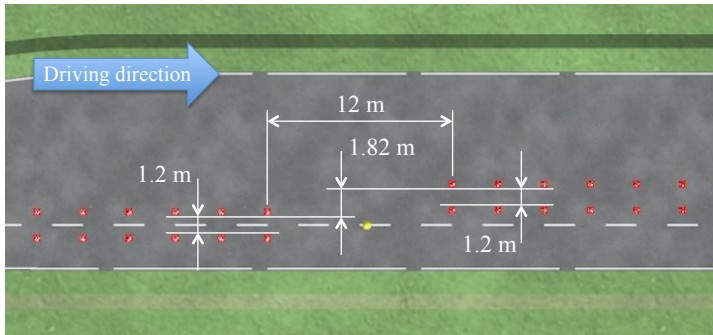

**Figure 20.** Single lane change course as obstacle avoidance.

In a closed loop, the driver recognizes the given lane change course in advance. Therefore, it makes it easier to pass the obstacle by steering a little to the opposite side before starting avoidance, as shown on the left in Figure 21. However, it is impossible to avoid oncoming obstacles in this way in the real world.

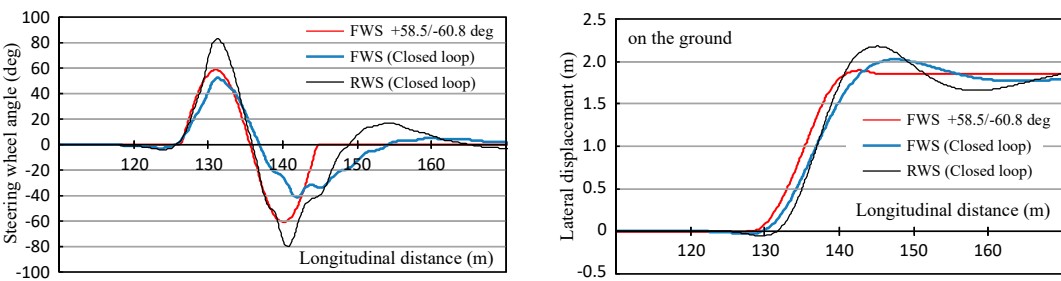

**Figure 21.** Steering wheel angle and lateral displacement.

As shown in Figure 21, in the case of FWS, the rise of the steering angle after starting avoidance is quite gentle, and the steering angle is considerably smaller and longer in the latter half of avoidance than in open loop. In the case of RWS, a large steering angle is required to recover from the delay of open loop avoidance. It is about a 1.6 times larger angle in the first half of avoidance and about twice as much in the second half. Overshoot at end of avoidance is also noticeable, indicating the difficulty of the avoidance operation in RWS.

### 4.1.4. Judgment of FWS and RWS on Implementation

The rear end of the RWS vehicle is pushed out at the first time of avoidance, and this causes a delay of lateral displacement. Superiority of FWS to RWS in obstacle avoidance performance was thus shown.

### 4.2. Comparison with Passenger Cars and Motorcycles

The obstacle avoidance capabilities of motorcycles which are accepted in the market are generally lower than those of passenger cars. When considering the social acceptability of completely new PMVs with tilting mechanisms, it is reasonable to compare them with those with market experience [7].

### 4.2.1. Vehicle Models of Passenger Cars and Motorcycles

CarMaker of IPG Automotive GmbH in Germany was used also as a passenger car simulation tool and an accompanying general passenger car model was used as a vehicle model. For the motorcycle, we used BikeSim from Mechanical Simulation, USA. Table 3 shows the basic specifications of these vehicles.

**Table 3.** Specifications of passenger car and motorcycle.

| Passenger Car | | | Motorcycle | | |
|---|---|---|---|---|---|
| Item | Unit | Value | Item | Unit | Value |
| Total length | m | 4.150 | Total length | m | 2.140 |
| Total width | m | 1.700 | Total width | m | 0.637 |
| Total height | m | 1.600 | Total height | m | 1.318 |
| Wheel base | m | 3.185 | Wheel base | m | 1.576 |
| Front tread | m | 1.508 | Front mass distribution | kg | 122.8 |
| Rear tread | m | 1.494 | Rear mass distribution | kg | 75.5 |
| Total mass | kg | 1463 | Total mass | kg | 198.3 |
| Gravity center height | m | 0.580 | Gravity center height | m | 0.350 |

### 4.2.2. Driving Course

The vehicle models drive into a single lane change course from the approach section. Course severity was set as maximum passing speed, which would be normal driving speed. Three levels, 1.5 m, 2.0 m and 2.5 m, of lateral displacement in a 10 m transition section were set by the parameter study in advance, as shown in Figure 22. The situation is that vehicles run at the right end of the lane and the obstacle is at the right end of lane. This means there is a severer situation for narrower vehicles, PMVs and motorcycles.

Generally, the lane width is set within a certain margin in relation to vehicle width. Therefore, course widths are not the same between PMVs and motorcycles, and passenger cars. However, lateral displacement levels are the same, and the capabilities of obstacle avoidance were judged on maximum passing speeds.

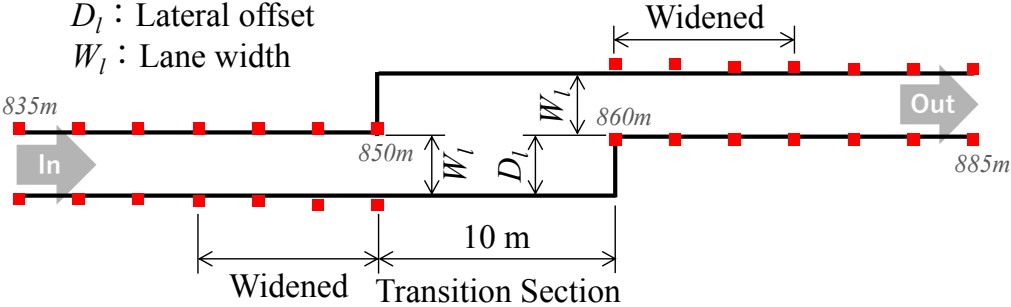

PMV, Motorcycle : $W_l$=1.5m  $D_l$=1.5m, 2.0m, 2.5m
Passenger Car      : $W_l$=2.32m  $D_l$=1.5m, 2.0m, 2.5m

**Figure 22.** Three levels of severity on driving course were set by parameter study.

### 4.2.3. Passing Speed Comparison and Understanding Social Acceptability

In the case of CarMaker, longitudinal speed is controlled consistently and automatically, but lateral control is switched from straight running model to automatic steering control model just before the transition section. The straight running model runs in the middle of the lane, but the automatic driving model takes the most advantageous course (feint motion) as far as possible, even if a narrow course width is set. For this reason, after entering a single lane change course, the model is switched from straight running model to automatic driving model. Switching timing was set as early so as not to miss the inner pylon and as late so as not to take the most advantageous course. Driver parameter and maneuver are shown in Table 4.

**Table 4.** Driver maneuvers were set to avoid feint motions.

| Driver | | | Maneuver | |
|---|---|---|---|---|
| **Item** | **Unit** | **Value** | **Item** | **Model** |
| cornering cutting coefficient (ccc) | | 0.1 | Longitudinal dynamics | IPGDriver |
| Max. longitudinal acceleration | m/s$^2$ | 3.0 | Lateral dynamics < 841.5 m | Follow course |
| Max. longitudinal deceleration | m/s$^2$ | -4.0 | ($D_l$ = 1.5 m) > 841.5 m | IPGDriver |
| Max. lateral acceleration | m/s$^2$ | 20 | Lateral dynamics < 843.0 m | Follow course |
| PylonShiftFdCoef | | 0.15 | ($D_l$ = 2.0 m) > 843.0 m | IPGDriver |
| | | | Lateral dynamics < 844.0m | Follow course |
| | | | ($D_l$ = 2.5 m) > 844.0 m | IPGDriver |

In PMVs and passenger cars, vehicle speed is automatically reduced when entering speed is quicker than passable speed. Therefore, in general, entrance speed (avoidance speed) to the transition section and exit speed from the transition section are different. In this study, entrance speed to the transition section was used as passing speed. In the case of PMV, only one occurrence of front inner wheel lifting was accepted and repeated liftings were judged to be failed.

In the case of BikeSim, autonomous maximum speed driving in the given course is not possible. A target trajectory was given to the rider model to drive a similar course to the PMV. The preview time was set according to the model switching distance in the PMV and the riding manner was adjusted by the behavior gain of tilting. The speed was gradually increased and it was judged from actual trajectory whether it was successful or not. The target trajectory was adjusted to get maximum passing speed in necessity.

A comparison of the maximum avoidable speeds is shown in Figure 23. As is generally understood, it was confirmed that there is a large difference between passenger cars and motorcycles. The capability of obstacle avoidance for PMVs is not only clearly superior to motorcycles, but is also equal to or higher than passenger cars. Therefore, we may understand that PMVs have sufficient social acceptability in terms of capability of obstacle avoidance.

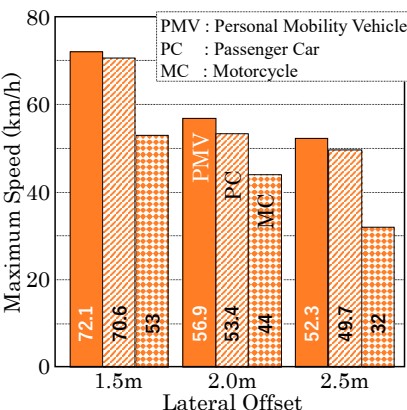

**Figure 23.** Comparison of capabilities on obstacle avoidance.

### 4.2.4. Mechanism of Superiority of PMV

As shown in Figure 24, for the dynamics of passenger cars, centrifugal force acting on gravity center height (G.C.H.) and supporting force acting on ground height cause roll moment, and lateral transfer of vertical load causes opposite the direction roll moment. These two moments are balanced with each other and the balance angle is expressed by Equation (7).

$$\tan \varphi_e \ = \ \tan \varphi_t \ = \ \frac{d_t}{\text{G.C.H.}} \tag{7}$$

$$\tan \varphi_e \ = \ \tan \varphi_a \ = \ \frac{d_a}{\text{G.C.H.}} \tag{8}$$

$\varphi_e$: equivariant roll angle
$d_t$: moment arm length on lateral transfer of load
G.C.H.: gravity center height
$\varphi_a$: actual roll angle
$d_a$: moment arm length on actual roll

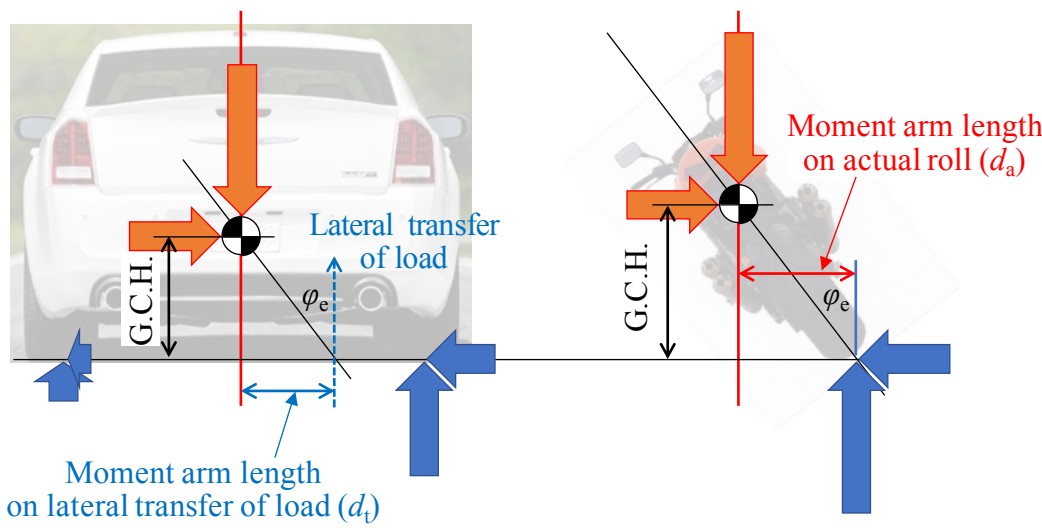

**Figure 24.** Roll moment balance on car and motorcycle.

In motorcycles, there is no lateral transfer of the vertical load. The roll moment due to the centrifugal force acting on the G.C.H. is balanced by the roll moment due to the combining of the vertical forces. The moment arm length moving in a lateral direction is caused by the actual vehicle roll angle. This is the balance angle of single-track vehicles such as bicycles and motorcycles and is expressed by Equation (8).

In PMVs with active tilting mechanisms, the strokes on both wheels are given to get the roll angle. This is equivalent to positively generating the lateral transfer of the vertical load. After the roll angle is obtained as equivariantly the same as motorcycles, the vertical loads of both wheels settle equally as a result. However, while the response of the roll angle is delayed, lateral transfer of the vertical load occurs between both wheels as in passenger cars.

As shown in Figure 25, the roll moment due to lateral transfer of the vertical load is the same as that of passenger cars, and the roll moment due to the actual roll angle of the vehicle body is the same as that of motorcycles. This is expressed by Equation (9). In the dynamics of passenger cars, lateral force and lateral acceleration on the front body occur without any delay due to the front tires slip angle by steering angle input. However, since the roll moment of inertia balances dynamically with the roll moment due to lateral acceleration, there is a slight delay in the lateral transfer of the vertical loads. In motorcycles, lateral acceleration can only be obtained when inward roll occurs in the vehicle body. This is a factor that causes a large delay in the lateral acceleration of motorcycles (slow avoidance motion), while delay in the lateral acceleration of a passenger car is not seen in avoiding obstacles.

$$\tan \varphi_e \;=\; \tan \varphi_a + \tan \varphi_t \;=\; \frac{d_a + d_t}{\text{G.C.H.}} \tag{9}$$

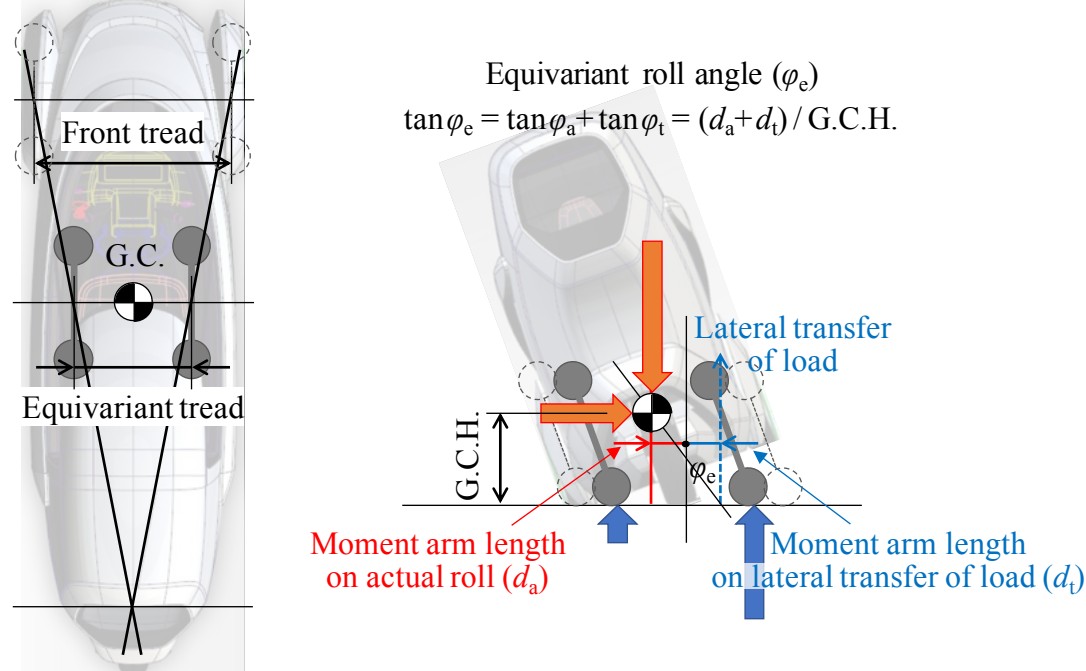

**Figure 25.** Roll moment balance on PMV with active tilting mechanism.

In the case of PMVs with an actively applied roll angle, as shown on the left of Figure 26, lateral transfer of vertical loads occurs with no delay according to the steering angle input. In comparison with motorcycles, it is equivalent to giving an additional virtual roll angle during delay of the actual roll angle by the tracking control and the roll moment of inertia. These two roll angles are shown on the right of Figure 26. The actual roll angle reaches its peak near the center of the transition section, but an additional virtual roll angle already balances with the lateral acceleration in the opposite direction at the same time.

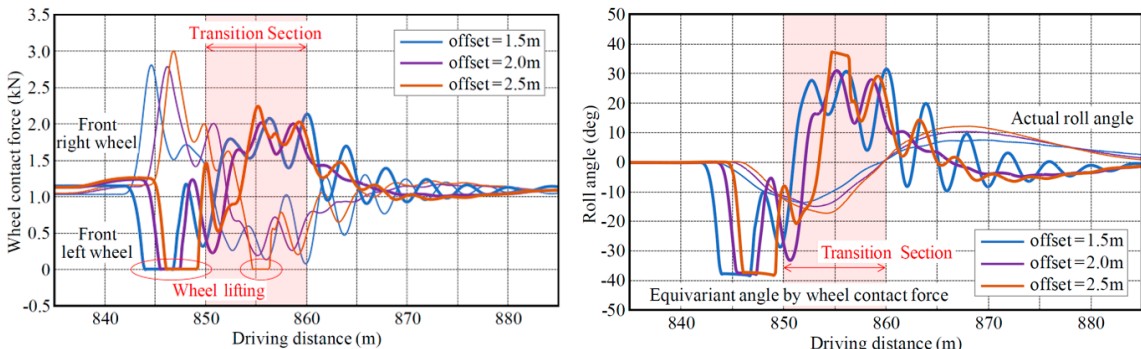

**Figure 26.** Lateral load transference and equivariant roll angle on load transference.

As shown in Figure 27, the sum of the actual roll angle of the vehicle body and the virtual roll angle of PMVs corresponds equivalently to the inward roll angle of motorcycles. Since this equivalent roll angle balances with lateral acceleration, PMVs are able to provide lateral acceleration without response delay, the same as with passenger cars.

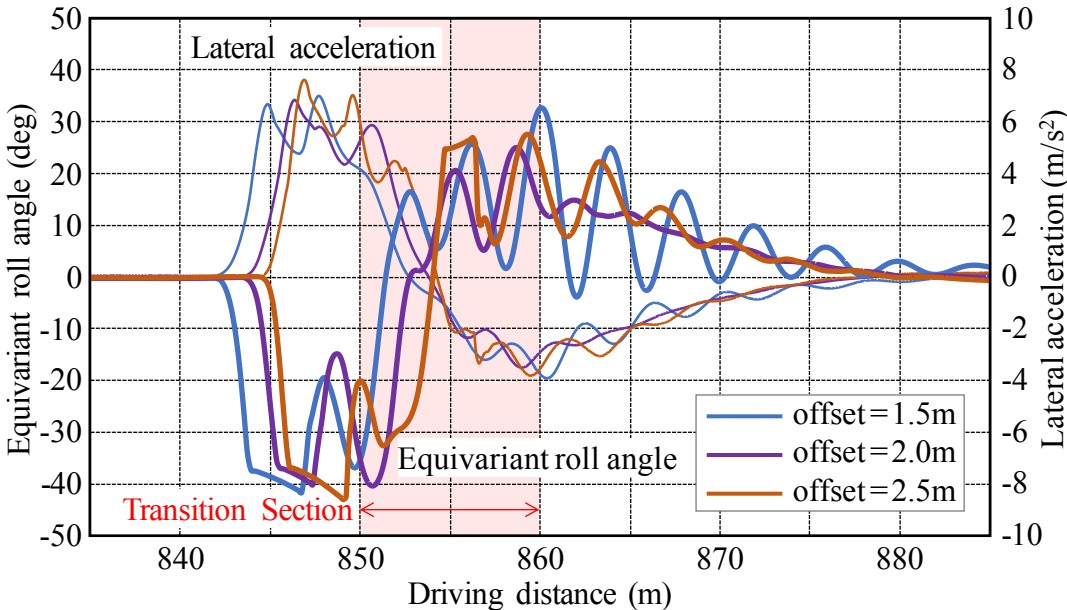

**Figure 27.** Balance of lateral acceleration and equivariant roll angle.

### 4.3. Summary of This Section

In a comparison of the capabilities of FWS and RWS on obstacle avoidance, it was observed that RWS has the behavior of the rear end being pushed out at the first timing and causes a delay of lateral displacement. Using dynamic simulation, it was shown to be especially difficult for the driver to steer to compensate for the delay and avoid the obstacle in RWS. Therefore, it can be judged that FWS is superior to RWS in obstacle avoidance performance.

PMVs with active tilting mechanisms have obstacle avoidance ability equal to or higher than that of passenger cars, because they have a much smaller roll moment of inertia than passenger cars and responsiveness equivalent to passenger cars. Thus, sufficient social acceptance of the capability of obstacle avoidance of PMVs with active tilting mechanisms could be confirmed also from their dynamic mechanism.

## 5. Energy Consumption on Active Tilting System

Although PMVs with a passive tilting mechanism have the negative point of a lack of self-standing ability in stopping and in very low speed the same as motorcycles, PMVs with an active tilting mechanism do not have such a concern. On the other hand, there is a concern about the contradiction with efficiency improvement inherent in PMVs, since energy consumption is inevitably unavoidable in active tilting mechanisms [8].

### 5.1. Mechanism of Energy Balance

#### 5.1.1. Energy Consumption in Active Tilting Mechanism

Although it is originally necessary to describe the twist energy of the stabilizer bar in relation to the sprung body independently of the bounce strokes of both front wheels, the consumed twist energy can be expressed by the product of the difference of the vertical loads from the mean value and the difference of the bounce strokes from the mean value, as shown in Equations (10) and (11) and Figure 28.

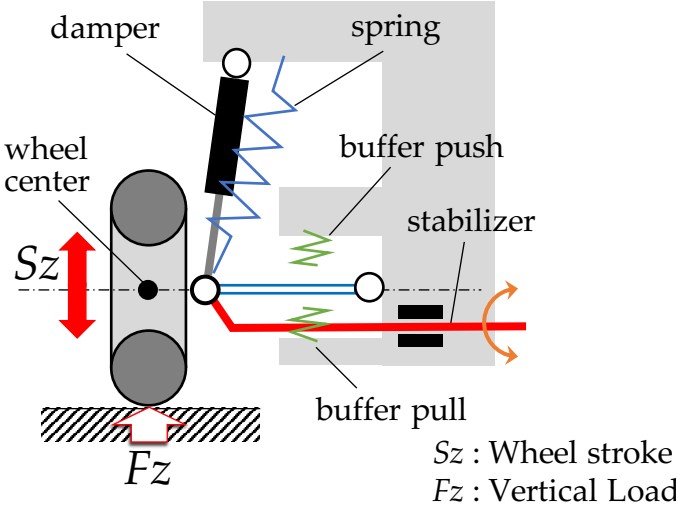

**Figure 28.** Energy consumption mechanism.

Energy efficiencies in actuating and recovering should be considered. In this study, the actuating efficiency was set to 0.5 and the recovering efficiency was set to 0.1 as an electrical general value.

$$\Delta E \ = \ F_{ZL} \times \Delta S_{ZL} + F_{ZR} \times \Delta S_{ZR} \tag{10}$$

$$\text{IF } \Delta E \geq 0, \ \Delta E^* \ = \ \Delta E \times 2 \ (\text{actuator energy efficiency rate } = \ 0.5)$$

$$\text{IF } \Delta E < 0, \ \Delta E^* \ = \ \Delta E \times 0.1 \ (\text{actuator energy efficiency rate } = \ 0.1)$$

$$E \ = \ \Sigma \, \Delta E^* \tag{11}$$

$E$: energy

### 5.1.2. Energy Consumption by Cornering Drag in Case of No Tilting

Generally, centripetal forces for vehicles are given as lateral force ($SF$) by tire slip angle ($SA$) when vehicles turn. Useful cornering force ($Y$) for turning is represented as $SF \times \cos{(SA)}$. Orthogonal $SF \times \sin{(SA)}$ is the cornering drag component, and energy is consumed wastefully.

Although the vertical load of each tire is always changing, for simplicity, the static load is substituted and the tire cornering power ($K$) is defined at this load, as shown in Figure 29. $Y$ is obtained from a dynamic simulation and $SA$ is obtained by dividing $Y$ by $K$.

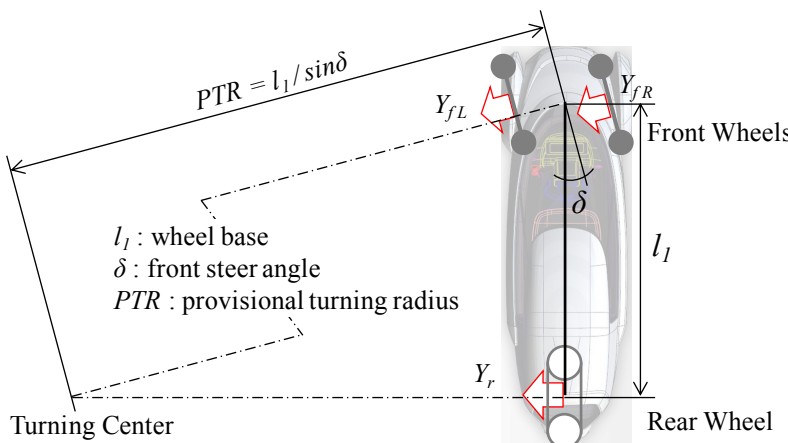

**Figure 29.** Energy consumption caused by cornering drag.

Total consumed energy rate caused by cornering drag is represented by the product of the cornering drag component and the longitudinal distance of the vehicle, as shown in Equation (12). Then, the accumulated consumed energy is the sum of the consumed energy rate, as shown in Equation (13).

$$\Delta E \ = \ Y_{fL} \times \sin\left(\frac{Y_{fL}}{K_{fL}}\right) \times v \times \Delta t + Y_{fR} \times \sin\left(\frac{Y_{fR}}{K_{fR}}\right) \times v \times \Delta t + Y_r \times \sin\left(\frac{Y_r}{K_r}\right) \times v \times \Delta t \tag{12}$$

$$E \ = \ \Sigma \, \Delta E \tag{13}$$

*E*: energy

*Y*: cornering force

*K*: cornering power

*v*: vehicle velocity

*t*: time

### 5.2. Energy Consumption in Typical Driving Modes

Although there are various explanations for the principle of side force generation due to the camber angle, it is usual that the cornering drag component of the side force by the camber angle is not considered, because it has no slip angle.

In this section, under the assumption that there is no cornering drag component of lateral force by camber angle, energy consumption for the active roll angle is compared with energy saving using the camber angle. Then, we consider the social acceptability of PMVs with active inward tilting mechanisms.

### 5.2.1. Energy Consumption on Steady Circle

In PMVs with inward tilting like motorcycles, during turning on the steady circle, the roll moment due to the centrifugal force is canceled by the roll angle and no additional roll moment is necessary. In other words, energy consumption due to cornering drag can continue to be avoided without energy consumption for tilting.

The energy consumption due to tilting and cornering drag of PMVs while entering the circle course, with a 50 m turning radius, from the straight course are shown. The vehicle starts on the straight course, accelerates while entering the circle course and reaches a steady speed of 60 km/h, as shown in Figure 30. Tilting energy is consumed only for a short time; therefore, the accumulated energy consumption for tilting may be considered essentially zero, as shown on the left of Figure 31. On the other hand, cornering drag energy is saved continuously on the circle course, as shown in right of Figure 31.

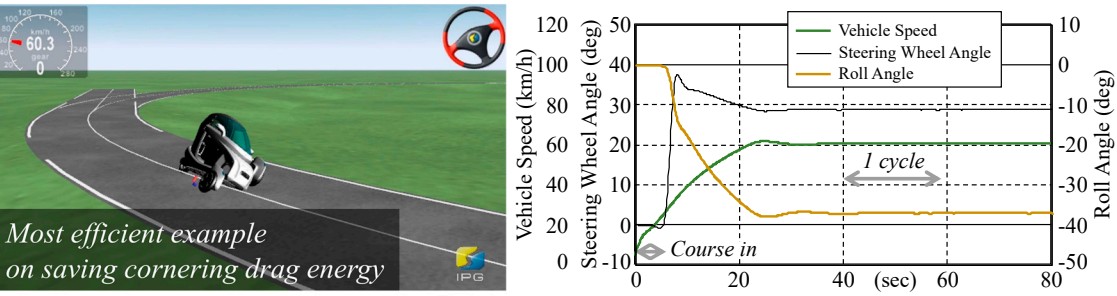

**Figure 30.** Driving on steady circle.

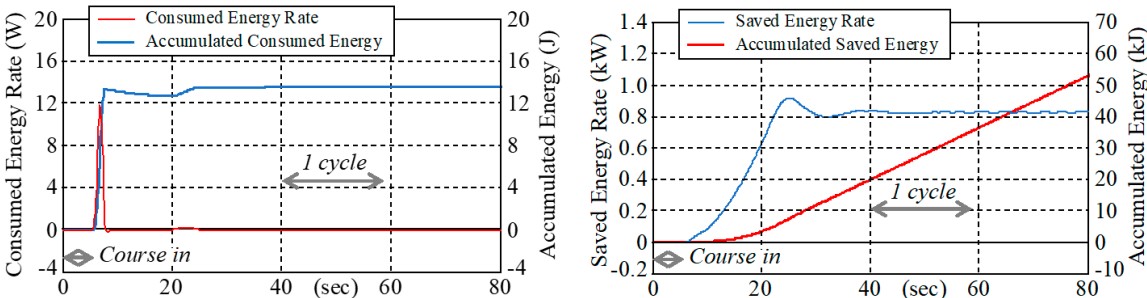

**Figure 31.** Consumed and saved energy on steady circle.

5.2.2. Energy Consumption in Slalom

In contrast to the steady circle, the pylon slalom with repeated tilting is a typical example of high energy consumption for tilting. We set the slalom course with 10 pylons at 18 m interval, as shown in Figure 32. The vehicle reaches 60 km/h before the slalom section and passes through the slalom section at a constant speed.

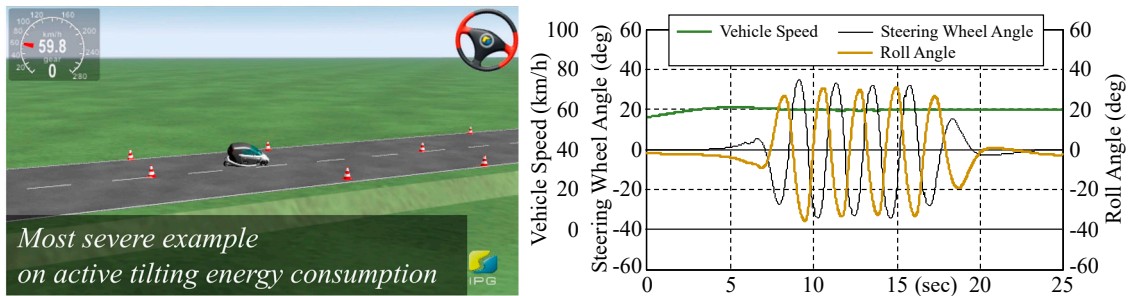

**Figure 32.** Driving on slalom course.

IPGDriver prepares the "corner cutting coefficient (ccc)" as one of the model parameters of driver characteristics in virtual driving. ccc = 0 means driver runs middle of given course and ccc = 1.0 means driver runs fully in-cut course. In other words, ccc = 1.0 is set to follow the easiest course. Under this condition, with ccc = 0.4, the driver could not pass the given course because of overturning. Therefore, this course setting may be considered to be almost at limit condition.

Although three levels of ccc were compared, which are ccc = 0.5, 0.7 and 0.9, as shown in Figure 33, no major difference was found in the steering wheel angles as input and lateral vehicle accelerations as output. Therefore, we decided to proceed with this study of energy balance at ccc = 0.7, which is considered to be a general driver characteristic.

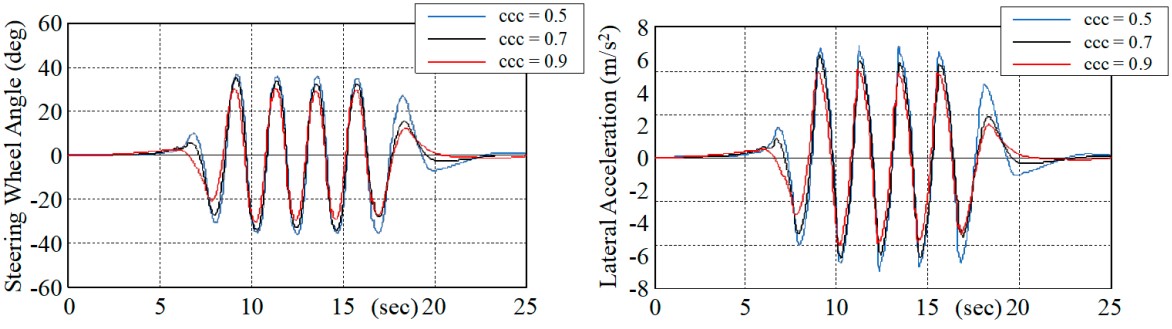

**Figure 33.** Influence of driver parameter is not significant.

In the slalom, a large amount of energy is consumed to give a quick change in roll angle, as shown on the left of Figure 34. During this time, although energy saving caused by avoidance of cornering

drag also occurs significantly, as shown on the right of Figure 34, this cannot offset energy consumption for tilting.

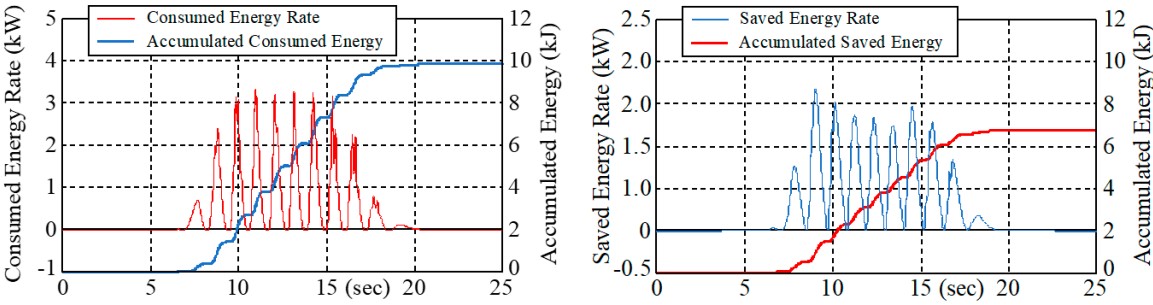

**Figure 34.** Consumed and saved energy on slalom course (ccc = 0.7).

### 5.3. Energy Consumption in Real World Condition

Driving modes for homologation of fuel consumption are basically on a straight course in all countries, and do not assume the steering wheel operation or lateral acceleration of the vehicle. However, German "auto motor und sport (AMS)" magazine prepares its own evaluation course, which is a round route on a public road in South Germany.

#### 5.3.1. Typical European Evaluation Course (AMS)

The AMS evaluation course with a total length of 92.5 km and elevation difference of 290 m is shown in Figure 35. It takes about 5000 s to drive this PMV. Maximum speed is determined by speed regulations and the limit of the vehicle's performance. There are many stop intersections on this course. The steering wheel angle is particularly large when a vehicle stops, starts and turns at these intersections.

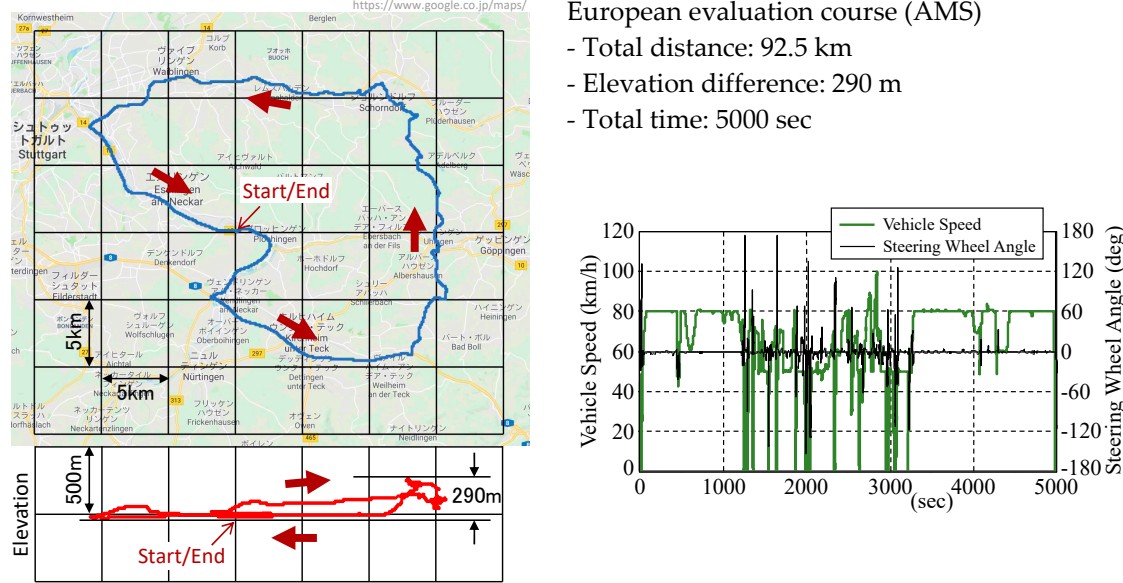

**Figure 35.** AMS evaluation course as typical European driving route.

#### 5.3.2. Energy Consumption in AMS Magazine Evaluation Course

As shown on the left of Figure 36, the energy consumption rate has sharp peaks at intersections and at tight corners, and the accumulated consumed energy is less than 6 kJ in 5000 s. This amount is fairly small.

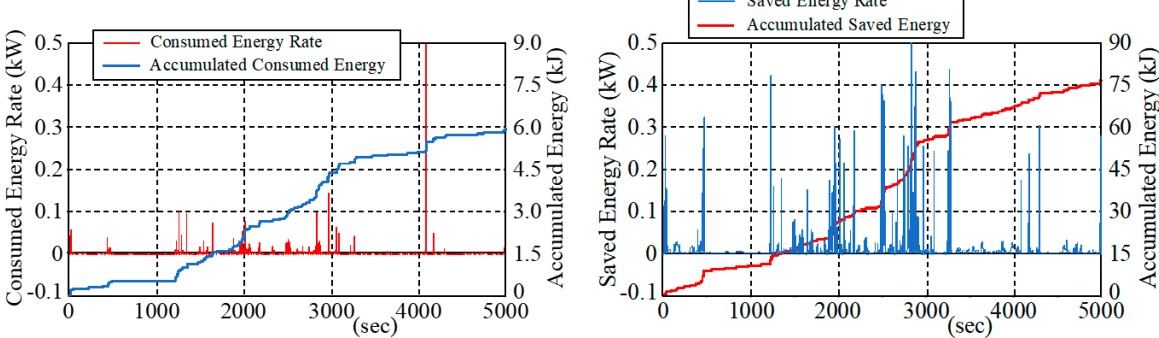

**Figure 36.** Consumed and saved energy on AMS evaluation course.

As shown on the right of Figure 36, energy saving caused by avoidance of cornering drag occurs much more frequently than energy consumption for tilting. This means that the steering wheel holding time on cornering sections is much longer than the steering input and return.

There is a difference in both energy rates and a larger difference in accumulated energy amounts. The accumulated energy saving against cornering drag reaches 76 kJ in 5000 s.

As shown in Figure 37, there is about 13 times difference between the two accumulated energies, and it shows no need to worry about energy consumption for active tilting mechanisms. High precision in the simulation model is not required at all for this definite result.

As shown in Table 2, the total mass of the vehicle including one passenger of this PMV is about 370 kg, and the total vertical load is about 3600 N. Assuming the tire rolling resistance coefficient (RRC) is approximately $80 \times 10^{-4}$, tire rolling resistance is approximately 29 N. With this rolling resistance, 2700 kJ of energy is consumed on the 92.5 km course.

This energy, 2700 kJ, is about 36 times the energy consumption by cornering drag and 450 times the energy consumption for the tilting mechanism, as shown in Figure 37. Energy consumption for driving PMVs is overwhelmingly dominated by tire rolling resistance if acceleration/deceleration and air drag are not considered.

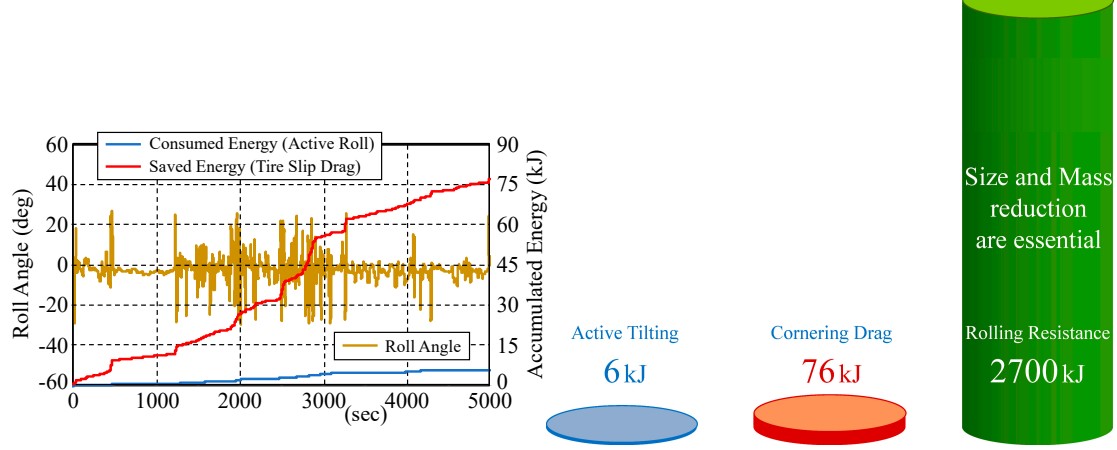

**Figure 37.** Energy balance on AMS course and comparison with rolling resistance energy.

*5.4. Social Acceptance from Viewpoint of Energy Consumption*

Energy consumption for tilting is only about one thirteenth of the saving effect of cornering drag; thus, the total efficiency of PMVs with an inward tilting mechanism is better than without a tilting mechanism. Additionally, smaller and lighter PMVs are significantly more efficient than general cars, as shown in Figure 38. These benefits of PMVs with an active inward tilting mechanism show sufficient social acceptability.

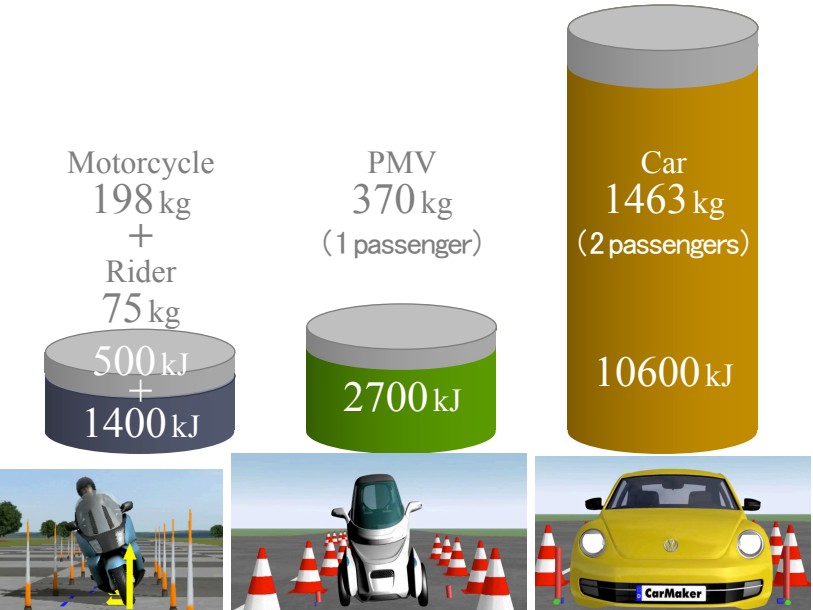

**Figure 38.** Comparison of energy consumption of general vehicles on rolling resistance.

## 6. Conclusions

PMVs with an active inward tilting mechanism with three wheels, double front wheels + single rear wheel and front steering + rear traction, are studied regarding front inner wheel lifting phenomena, capability of obstacle avoidance and energy balance of active tilting mechanism.

From the following results, it was shown that PMVs with an active inward tilting mechanism have sufficient social acceptability.

- Front inner wheel lifting phenomenon is only that vehicle roll is unavailable to follow target roll angle *TRA* due to roll moment of inertia of sprung mass $Ixx^*$. Smaller $Ixx^*$ and larger $K\varphi$ improve front inner wheel lifting phenomena.
- It is possible to suppress unstable roll phenomena without intentional steering input related to front inner wheel lifting phenomena by decreasing the I gain of roll tracking control (PID).
- PMVs with an active tilting mechanism have two advantages of vehicle dynamics. One is lateral transfer of vertical loads on front both wheels, in the same way as general cars, and the other is inward tilting on turning, the same as motorcycles.
- PMVs with an active tilting mechanism have an obstacle avoidance ability equal to or higher than that of passenger cars, because they have a much smaller roll moment of inertia than passenger cars and responsiveness equivalent to passenger cars.
- Energy consumption for an active tilting mechanism is much smaller than energy saving due to avoiding cornering drag by using the camber angle. This shows there is no need to worry about energy consumption with active tilting mechanisms.
- From a general energy efficiency point of view, smaller and lighter PMVs are significantly more efficient than general cars, because rolling resistance caused by vehicle mass and the tire rolling resistance coefficient is the major resistance in these vehicles.

**Author Contributions:** Conceptualization, T.H.; formal analysis, T.H.; methodology, T.H. and T.K.; supervision, I.K.; validation, T.H.; writing—original draft, T.H.; writing—review and editing, T.H.

**Funding:** This research received no external funding.

**Conflicts of Interest:** The authors declare no conflict of interest.

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
