# Peer review of "Study of Personal Mobility Vehicle (PMV) with Active Inward Tilting Mechanism on Obstacle Avoidance and Energy Efficiency"

_applsci, doi:10.3390/app9224737_

Round 1

Reviewer 1 Report

The article is well written and interesting to read. However, some minor revisions should be conducted:

The results of this study should be reported in abstract. Add a reference for CarMaker [Line 90]. Use more relevant references. Check the journal standard format for equations etc. Please, highlight the roles of road geometry on the performance of Personal Mobility Vehicle (PMV). Clarify the aim of this research and its contribution to the science. Does this article overlap with the following article? If so please explain (%)?

Haraguchi, Tetsunori, Ichiro Kageyama, and Tetsuya Kaneko. "Inner Wheel Lifting Characteristics of Tilting Type Personal Mobility Vehicle by Sudden Steering Input." Transactions of Society of Automotive Engineers of Japan 50.1 (2019).‏

https://www.jstage.jst.go.jp/article/jsaeronbun/50/1/50_20194038/_pdf

Improve the writing skills.

Author Response

Thank you very much for your kind reviewing

> The results of this study should be reported in abstract. Res; Abstract was modified on your comment.

> Add a reference for CarMaker [Line 90].
Res; Online URL was referred, and article was modified.

> Check the journal standard format for equations etc.
Res; My recent version of MS Word can not follow the journal standard format. I could not insert equations on the journal template because of the deference of the versions of MS Word. I included equations as text on recent version additionally, but font is not match to the journal standard. I hope editorial office manage to modify.

> Please, highlight the roles of road geometry on the performance of Personal Mobility Vehicle (PMV).
Res; Sorry, I do not understand this comment well.

> Clarify the aim of this research and its contribution to the science.
Res; I understand this comment is related with the comment for abstract. I modified abstract and modify Line 40-41 in order to clarify the aim of this research.

> Does this article overlap with the following article? If so please explain (%)?
Res; Yes, it is. The former article was written in Japanese. My manuscript includes story from my Japanese article as a part of story, and is written in English totally from higher point of view than former Japanese article. JSAE permits on its regulation for original authors to reuse his/her original figures, etc. on the other articles.

Reviewer 2 Report

The paper titled “Study of Personal Mobility Vehicle (PMV) with active inward tilting mechanism on obstacle avoidance and on energy efficiency” investigated the use of an alternative way of transport named Personal Mobility Vehicle (PMV) based on a three-wheeled vehicle. With this study, the authors investigated the optimum kinematic solution for minimizing problems concerning wheel lifting phenomena. The authors also investigate on influence of the geometry parameters of the vehicle and the influence of PID factors on roll angle tracking control.

The results reported, underline the potential of multibody dynamic simulators for evaluating the behavior of new systems and vehicles for better designs. The paper is well-written, concise, and easy to understand. It would be nice to know if geometrical instabilities have been taken into account during the design phase of the model.

Author Response

Thank you very much for your kind reviewing.
I modified the original article.